# Inosine Pranobex Significantly Decreased the Case-Fatality Rate among PCR Positive Elderly with SARS-CoV-2 at Three Nursing Homes in the Czech Republic

**DOI:** 10.3390/pathogens9121055

**Published:** 2020-12-16

**Authors:** Jiří Beran, Marian Špajdel, Věra Katzerová, Alena Holoušová, Jan Malyš, Jana Finger Rousková, Jiří Slíva

**Affiliations:** 1Department for Tropical, Travel Medicine and Immunization, Institute of Postgraduate Health Education, 100 05 Prague, Czech Republic; 2Department of Psychology, Faculty of Philosophy and Arts, Trnava University, 918 43 Trnava, Slovakia; marian.spajdel@truni.sk; 3Domov Důchodců Litovel, 784 01 Litovel, Czech Republic; katzerova@volny.cz; 4Sanatorium Topas, 534 01 Holice, Czech Republic; alena.holousova@sanatorium-topas.cz (A.H.); jan.malys@sanatorium-topas.cz (J.M.); 5Senior dům Beránek Úpice, 542 32 Úpice, Czech Republic; rouskova.jana@email.cz; 6Department of Pharmacology, Third Faculty of Medicine, Charles University, 100 00 Prague, Czech Republic; slivaj@seznam.cz

**Keywords:** inosine pranobex, natural killer cells, acute respiratory viral infection, treatment, COVID-19, nursing homes, case-fatality rate

## Abstract

During the COVID-19 pandemic, the elderly population has been disproportionately affected, especially those in nursing homes (NH). Inosine pranobex (IP) has been previously demonstrated to be effective in treating acute viral respiratory infections. In three NH experiencing the SARS-CoV-2 virus epidemic, we started treatment with IP as soon as clients tested PCR+. In Litovel, CZ, the difference in case-fatality rate (CFR) for the PCR+ group using vs. not using IP was statistically significant, and the odds ratio (OR) was 7.2. When comparing all those taking IP in the three NH vs. the non-drug PCR+ group in Litovel, the odds ratio was lower for all three NH, but still significant at 2.9. The CFR in all three tested NHs, age range 75–84, compared to the CFR in all NHs in the Czech Republic, was significantly reduced (7.5% vs. 18%) (OR: 2.8); there was also a significant difference across all age groups (OR: 1.7). In our study with 301 residents, the CFR was significantly reduced (OR: 2.8) to 11.9% (17/142) in comparison to a study in Ireland with 27.6% (211/764). We think the effect of IP was significant in this reduction; nevertheless, these are preliminary results that need larger-scale trials on COVID-19 patients.

## 1. Introduction

The pandemic caused by severe acute respiratory syndrome coronavirus 2 (SARS-CoV-2) was first detected in Wuhan in December 2019 and has since spread around the globe. Real-time PCR assays are recommended for diagnosing the SARS-CoV-2 infection [1]. Clinical presentations include mild, moderate, or severe cases, which are easily counted. However, asymptomatic infections are hidden from statistics.

Viral dynamics, cellular immunity, and the antibody response in infected but asymptomatic patients are still not fully understood [2,3]. The initial infectious dose significantly influences the length of the incubation period and the clinical presentation; as a result, we can see in any given population coronavirus disease (COVID-19) cases ranging from mild to severe. Very low infectious doses can be a reason for asymptomatic infections, but it is very difficult to demonstrate this at the population-level with serological tests [2,3].

Any successful control of a viral infection involves the complex interplay between diverse cell types associated with both innate and adaptive immunity. Natural killer (NK) cells are a type of innate lymphoid cell that plays an important role in the first line of immune defense against any viral infection, including COVID-19. They constitute the primary rapid, innate immune attack on virus-infected cells [4] (Figure 1).

NK cells kill infected cells rapidly and directly, without antigen presentation or recognition. In response to stimuli from diverse sources including infections, cytokines, stress, or other immune cells, NK cells exert the following distinct actions: (1) secretion of perforin and granzyme to directly kill target cells, (2) release of cytokines to regulate the immune response, and (3) coupling with death-inducing receptors on target cells, which induce apoptosis [5,6,7].

Adaptive immunity is the second line of antiviral defense and is based on antigen presentation. The virus is processed by antigen presentation cells (APC), and the most important antigens are displayed on the APC surface together with molecules of the major histocompatibility complex (MHC) [8].

In the immune system, there is an equilibrium between Th1 (T-helper 1) and Th2 (T-helper 2) cell activity. When Th1 activity is increased, as part of cellular immunity, Th1 cytokines are produced that suppress Th2 activity and vice versa. Immune system activity is similar to the pendulum of a clock that swings from one side (Th1 stimulated by thymus-dependent intracellular antigen) to the other (Th2—thymus dependent extracellular antigen) depending on the stimulus and type of antigen. You cannot have both actions at the same time.

When a virus replicates inside a target cell, it is considered by the immune system as a thymus-dependent intracellular antigen, and thus, adaptive immunity via Th1 and cytotoxic T-lymphocytes (CTL) is employed as the second line of the body’s anti-viral immunity. Additionally, cytokines IL-2, TNF-α, and INF-γ, which activate macrophages, T-cells, and dendritic cells, are released. In terms of cytotoxic activity, CTLs have a role very similar to NK cells [8,9].

When these two lines of immunity are overwhelmed, the virus is released into the blood, which leads to viremia. The virus is now mostly extracellular and thymus-dependent extracellular antigen and Th2, in co-operation with T follicular helper cells (Tfh), begin to stimulate antibody production. Antibodies then bind to extracellular antigens on the virus particles and mark them for destruction by macrophages.

The elderly population has been particularly and severely affected by SARS-CoV-2 [10,11], which may be potentially explained by immunosenescence, malnutrition, comorbidities, polypharmacotherapy, and inflammatory conditions in these patients [12,13]. As a result, SARS-CoV-2 has disproportionately affected the residents of nursing homes (NH). For instance, in Ireland, they performed a survey of NHs. Surveys were returned from 62.2% (28/45) of the surveyed NHs (2043 residents, 2303 beds), and three-quarters (21/28) reported COVID-19 outbreaks (1741 residents, 1972 beds). The incidence among residents was 43.9% (764/1741): 40.8% (710/1741) laboratory-confirmed, with 27.2% (193/710) asymptomatic, and 3.1% (54/1741) as clinically suspected cases. Case-fatalities among residents were 27.6% (211/764). This study demonstrates the significant impact of COVID-19 on the NH sector. Systematic point-prevalence testing was suggested to reduce the risk of transmission from asymptomatic carriers and help manage outbreaks in NH settings [10].

Inosine pranobex (IP), also known as inosine acedoben dimepranol, inosine pranobex, and methisoprinol, is a synthetic compound with immunomodulatory and antiviral properties. The drug was initially authorized in 1971, and it is currently marketed in more than 70 countries for treating viral diseases, including subacute sclerosing panencephalitis, acute viral respiratory infections, measles, herpes simplex infections, varicella, human papillomavirus, cytomegalovirus, and Epstein–Barr virus [14].

Clinical and immunological studies conducted over the past five years have confirmed the effect of inosine pranobex (IP), via natural killer (NK) cells and cytotoxicity, for treating the majority of investigated viral infections; this efficiency will hopefully be transferrable to the currently spreading acute viral respiratory infection COVID-19 [4].

Studies on IP have demonstrated that its immunomodulatory activity is characterized by enhancing lymphocyte proliferation, cytokine production, and NK cell cytotoxicity [14,15]. The effects of IP at the cellular level have been investigated in detail [15], and it was found that NK cells exposed to IP had increased expression of multiple NKG2D ligands, leading to increased NKG2D-dependent target cell immunogenicity. In a recent study [16], IP administration was shown to elicit early and consistent increases in NK cell levels within 1.5 h of receiving IP and with two-fold or higher level by the fifth day. IP-induced NK populations contained granzyme A and perforin [16,17].

IP is a potent medication for treating or preventing viral infections under various conditions, including patients who are elderly but otherwise healthy. This is particularly important since, in these populations, NK performance is known to be compromised, and insufficient performance may be a key contributor to high rates of viral infections associated with immunosenescence [18,19,20].

The aim of this prospective trial was to critically assess the possible benefits of IP in the prevention and treatment of COVID-19 in nursing home settings.

## 2. Results

### 2.1. Patients Characteristics

Starting in June 2020, the COVID-19 epidemic began devastating the nursing home in Litovel, CZ. Out of a total of 56 residents, 33 tested positive. Of these, 19 began to show clinical signs and were started on inosine pranobex, at a dose of two tablets three times a day for 7 days. Patients with uricemia or on dialysis took a dose of one tablet three times a day for 1 day only. Of the patients taking inosine pranobex, five patients were hospitalized for pneumonia in the department of infectious diseases. All but one patient received two treatments with anti-COVID plasma; no antiviral treatments were administered while the patients were in the infectious disease department. They continued with symptomatic treatment and started, when necessary, antibiotic treatment, mineral supplementation, rehydration, and oxygenation.

The COVID-19 epidemic also hit a second NH in Sanatorium Topas Holice, CZ, during the second week of September. At that time, the home had a capacity of 174 people. A total of 86 tested positive for SARS-CoV-2 and began taking inosine pranobex at a dose of two tablets, three times a day for 7 days. COVID negative residents (88) used inosine pranobex preventively two tablets once a day for 10 days.

The epidemic appeared in a third NH, in Beránek Úpice, CZ, at the beginning of September. The capacity of the home was 71 people, and 37 tested PCR+ on four separate test dates spread over a 20 day period. These residents started taking inosine pranobex at a dose of two tablets, three times a day for 7 days. Nothing was given to clients with a negative PCR test.

In Litovel, inosine pranobex was administered to PCR+ residents only after they began to show symptoms of COVID-19. In the other two nursing homes, they started giving inosine pranobex to all clients immediately after testing positive (PCR); as such, there was no control group. Table 1 shows the data clearly.

### 2.2. Recorded Case-Fatality Rate

In Litovel, none of the hospitalized patients died but one PCR+ NH resident receiving inosine pranobex died. It is noteworthy that he was not treated with inosine pranobex until 6 days after testing positive. The patient was an otherwise healthy 95-year-old man who was PCR+ for SARS-CoV-2, with no symptoms of COVID-19 from the date of his PCR test until the sixth day. He then experienced acute onset of weakness and developed difficulty breathing (SpO_2_ 92%); treatment with 3 g of IP was initiated; death came suddenly 5 days later.

In the other Litovel group, of the 14 positive patients not taking inosine pranobex, four patients died. The difference in the number of deaths was statistically significant, and the odds ratio (OR) was 7.2 (95% CI: 0.71–73.54; *p* = 0.0324). This indicates that a PCR+ nursing home client taking inosine pranobex is seven times less likely die than one not taking IP. The difference in the case-fatality rate (CFR) of PCR+ residents between the group using and not using inosine pranobex was statistically significant.

The patients who died were very old (73, 90, 92, and 93 years). They did not develop typical respiratory symptoms of COVID-19. Instead, they typically experienced sudden onset weakness, low oxygen saturation (SpO_2_), or suspected micro-embolization. Importantly, all were polymorbid and none were treated with IP. Patient deaths were attributed to gastrointestinal problems (i.e., icterus, ileus, and gastroenteritis with severe electrolyte imbalances).

In the Sanatorium Topas Holice NH, a total of 11 PCR+ residents treated with inosine pranobex died. Among these taking inosine pranobex prophylactically, there were no new PCR+ diagnoses among the residents.

In the Beránek Úpice NH, a total of five (age 88–95) PCR+ residents died while on inosine pranobex.

These results are shown in Figure 2 and Figure 3. Inosine pranobex had a significant effect on the course of the disease in institutionalized patients older than 65 years of age. When comparing the CFR of all those taking inosine pranobex in the three nursing homes with the non-IP PCR+ group in Litovel, the difference was still statistically significant. The odds ratio (2.9) was lower for all three nursing homes but still significant (95% CI: 0.8–10.3), i.e., the chances of survival were still three times higher for residents taking inosine pranobex. The decrease in survival odds was also potentially due to the heterogeneity of the residents in the three nursing homes. Two of the homes were Alzheimer’s centers (Sanatorium Topas Holice and Beránek Úpice), and the third (Litovel) was a very large, specialized facility (174 residents), which had a building layout that facilitated the spread of COVID because it was all but impossible to totally isolate positive residents from the rest of the nursing home’s population. This was why IP was used prophylactically with the PCR negative residents in this NH.

The age cohort of 75–84 year-olds was significantly influenced in all three nursing homes, where the reduced CFR compared to the CFR in other nursing homes in the Czech Republic was statistically significant—there was a reduction in CFR from 18% to 7.5% in the specified age group within the selected homes. The odds ratio was 2.8 (95% CI: 0.8–9.6; *p* = 0.047). Nevertheless, there was also a significant difference across all age groups—OR: 1.7 (95% CI: 0.96–2.98; *p* = 0.031).

## 3. Discussion

Acute respiratory infections are globally the most common type of viral infections. Severe forms are responsible for approximately 3.9 million deaths per year and are one of the leading causes of morbidity and mortality worldwide [21]. This number is likely to increase considerably with the current spread of COVID-19.

In view of the current COVID-19 pandemic, it is important to consider the results of studies using clinical subjects with laboratory-confirmed acute viral respiratory infections, which have been conducted to compare the efficacy and safety of specific treatments, in this case, IP, compared to placebos [22]. In this study, the primary efficacy endpoint was considered the “time to resolution of all influenza-like disease-associated symptoms.” IP 500 mg tablets or placebo were self-administered orally for 7 days (i.e., two tablets, three times per day). The first dose was taken immediately after case randomization at the clinic, and the remaining doses were self-administered at the nursing home. The medication was taken at intervals of approximately 8 h but was adjusted to fit the patient’s lifestyle so as not to interfere with normal sleeping patterns. Each patient received a kit containing either IP or placebos sufficient for 7 days of treatment [22]. In subjects less than 50 years of age and without any associated on-going disease, statistically significant differences in “time to resolution of all influenza-like disease-associated symptoms” were observed between treatment groups, and patients treated with IP showed faster improvement than those treated with placebo [22].

A different study investigated the breadth and kinetics of the immune response during a non-severe case of COVID-19 [23]. The study provided valuable insights in that it paralleled the immunological results and clinical conditions of the diseased in a previously healthy 45 year-old female patient from Wuhan, China, who traveled to Melbourne, Australia. The patient did not experience respiratory failure or acute respiratory distress syndrome; she did not require supplemental oxygenation and was discharged after one week of hospitalization, which indicates she had a non-severe but symptomatic course of the disease. Clinical findings in the lungs were observed on day five of hospitalization; however, these resolved after 10 days. It is important to consider the time sequence of the body’s immune response: first, the activity of minimally active NK cells is enhanced; however, when their activity decreases, cytotoxic T lymphocytes (containing granzymes A and B and perforin) begin to increase; the levels of helper T lymphocytes are maintained during this process, which is then followed by an increase in antibody-secreting cells (ASCs) together with T follicular helper (Tfh) cells. This leads to an increase in the production of antibodies, with maximum levels of IgG and IgM occurring on day 20. When lymphocyte cytotoxic activity is induced, the course of COVID-19 is often mild, and clinical findings in the lungs disappear within a few days even without treatment. In such cases, it is probably not necessary to stimulate the immune responses with IP [23]. Nevertheless, in elderly patients or in those who are suffering from pre-existing chronic diseases, IP treatments have the potential to substantially increase the levels of NK cells, accelerate the resolution of symptoms, and can prevent a potential decline into primary viral pneumonia.

The results of previous studies indicate that IP can boost the levels of phenotypically competent NK cells in healthy individuals during conditions associated with acute viral respiratory infections, which indicates that it can be used to improve potentially compromised immune functions [16,22].

In Ireland, in a large study involving 28 nursing homes (2043 residents, 2303 beds) during the COVID-19 outbreak, 21 institutions were affected (1741 residents, 1972 beds) with 43.9% of the residents infected with COVID-19. The resident case-fatality was 27.6% (211/764) [10]. Recently, data with detailed demographics from five care homes in Northern Ireland were published [24]. In total, 388 individuals were tested (245 residents/87 positive, and 143 staff/10 positive). Most of the tested residents were women (72%). Those testing positive had a mean age of 86.4 years (SD, 8.05) and CFR was 31%. With regard to other international comparisons on NH COVID-19 infection rates and case-fatalities, results are consistent with a report on eighty-nine residents in a US facility [25], with an infection rate of 64% and a CFR of 26%, and a series from 394 residents in four UK NHs with an infection rate of 40%, and a CFR of 26% [26].

Our study consisted of three nursing homes with 301 residents, 156 of whom (51.8%) tested positive for the SARS-CoV-2 virus (PCR test). This was a higher rate than in the nursing homes in Ireland, but resident case-fatality was much lower (11.9% (17/142)), which can probably be attributed to inosine pranobex. The average age of residents who tested PCR positive at the three NHs (Litovel, Holice, Úpice) was 82.7, 84.9, and 85.5 years with median age 83.5, 86.9, and 86.0 years, respectively. Most of them were women (68.4%, 72.1%, 89.2%, and for all three NHs 76.1%). These demographic data are very similar to other publications [24].

In our study, we demonstrated the positive effects of IP on nursing home residents, which is clearly the population at the highest risk for a severe course of COVID-19. The CFR in all three nursing homes, as of the end of the first epidemic wave, in those over 65 years of age, was lower than in the residents of similar nursing homes elsewhere in the Czech Republic, and this difference was statistically significant OR: 1.7 (95% CI: 0.96–2.98; *p* = 0.031). The CFR was also lower than in the residents of similar types of nursing homes in Ireland (OR: 2.8 (95% CI: 1.6–4.8; *p* = 0.00008). The benefit of IP was also significant in the cohort of those 75–84 years of age. This is probably because, in this particular group, while the immune system might be somewhat compromised, it is still capable of responding positively to the effect of IP. It is critically important to start IP (after considering contraindications) in those over 65 years, as soon as they test PCR+ for COVID-19. Evidence suggests that the case-fatality rate and the number of hospitalizations can be significantly reduced with this approach. Nevertheless, these are preliminary results that need confirmation using larger-scale trials on COVID-19 patients.

To conclude, IP (as an off-label drug in some countries) can be used for treating COVID-19 in at least some cases of infection and to potentially minimize the severity of the disease course.

In elderly patients who are immunosenescent (especially residents of nursing homes) and have been exposed to SARS-CoV-2, initial treatments should be a maximum dosage of 1 g (two tablets) per 10 kg of body weight three or four times per day for 7–10 days or until 2 days after symptom resolution.

## 4. Materials and Methods

### 4.1. Subjects

This prospective clinical study was performed in three nursing homes in three towns (Litovel, Sanatorium Topas Holice, and Beránek Úpice) in the Czech Republic. The study ran from June 2020 until September 2020. While the first home was an unspecialized nursing home, the other two specialize in caring for clients with Alzheimer’s disease. The number of clients, average age (AG), median age (MA), and percentage of women (PW) of PCR positive residents for the three selected homes were as follows (1) Litovel NH: 67, AG = 82.7 years, MA = 83.5 years, PW = 68.4%, (2) Sanatorium Topas Holice NH: 174, AG = 84.9 years, MA = 86.9 years, PW = 72.1%, and (3) Beránek Úpice NH: 71 AG = 85.5 years, MA: 86.0 years, PW = 91.2%, respectively. The PCR-testing for the SARS-CoV-2 virus was performed on all clients.

### 4.2. Treatment

All PCR+ clients for SARS-CoV-2 were treated with inosine pranobex as soon as a positive test was recorded. Isoprinosine 500 mg tablets (each tablet contains 500 mg of inosine pranobex; Ewopharma International, s.r.o., Prokopa Veľkého 52, 811 04 Bratislava, Slovak Republic) was administered to patients; however, it was not used precisely as it is currently described in the summary of product characteristics (SmPC), i.e., based on the patient’s body weight and the nature and severity of the disease. The dosage regimen was based on practical limitations, i.e., elderly patients 65 years and older: received two tablets 3 times a day. The average duration of treatment was seven days.

Clients who were PCR negative for COVID (in Topas Holice) started taking isoprinosine 500 mg tablets at a dose of 2 tablets once a day for 10 days as soon as the infection was detected within the nursing home.

At all three NHs, PCR+ patients received symptomatic treatment and, if necessary, also mineral supplementation, rehydration, and oxygenation. If residents developed signs of bacterial pneumonia (i.e., C-reactive protein > 70 mg/L, auscultation findings, fever, low SpO_2_), then antibiotic treatment was initiated.

Altogether there were 21 patients hospitalized at various infectious diseases departments. Except for one patient from the Litovel NH, none of them requested specific antiviral (anti-COVID-19) treatment during hospitalization (i.e., remdesivir or anti-COVID plasma); however, intensive antibiotic treatment, mineral supplementation, and rehydration was provided as needed.

### 4.3. Statistical Analysis

All analyses were conducted using SPSS^®^ software version 20 (IBM, New York, NY, USA). The chi-square test was used to study associations between two categorical variables (one-sided hypothesis) with a 5% level of significance. No adjustments for multiplicity were made. The odds ratios were calculated with 95% confidence intervals (CIs). All analyses grouped subjects according to the treatment they actually received.

### 4.4. Ethical Statement

The study was conducted in accordance with the Declaration of Helsinki, and the project was approved by the ethics committee of the University Hospital Hradec Králové on 19NOV2020 (project identification code ISO-ELD-2020 and Ethics Committee (EC) approval no.: 202012 P05).

## Figures and Tables

**Figure 1 pathogens-09-01055-f001:**
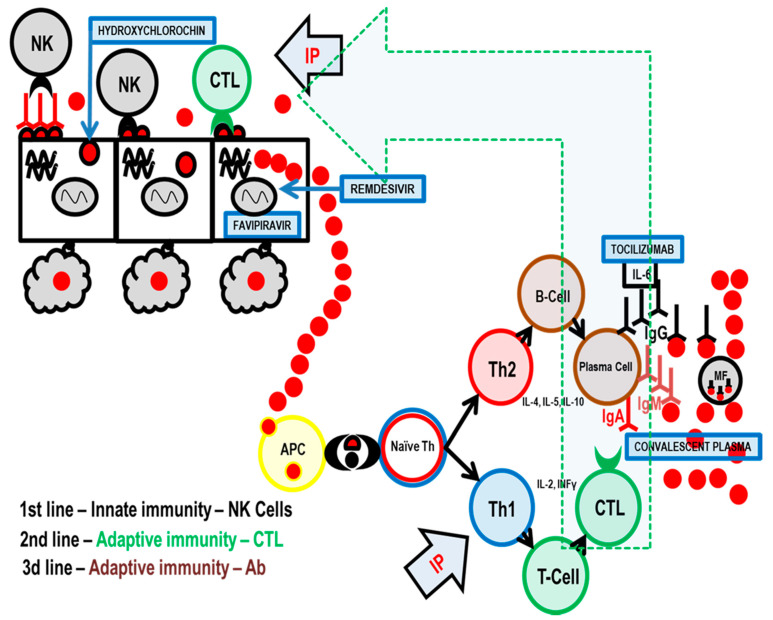
Three levels of antiviral immunity and the influence of specific COVID-19 treatment options. Legend: red dot—virus SARS-CoV-2; IP—inosine pranobex; NK cells—natural killer cells; CTL—cytotoxic T-lymphocyte (CD8+); APC—antigen presenting cell; Naïve Th—naïve helper T lymphocyte (CD4+ cell); Th1—type of helper cells that lead to an increased cell-mediated immunity; Th2—type of helper cells that lead to a humoral immune (antibody) response; T-Cell—a type of lymphocyte, which differentiate into helper, regulatory, or cytotoxic T cells or become memory T cells; B-Cell—B cells are involved in humoral immunity and differentiate into a plasma cell; plasma cell—short-lived antibody-producing cell derived from B-Cell; IgA, IgM, IgG—antibody classes of immunoglobulins; MF—macrophages, specialized cells involved in the detection, phagocytosis and destruction of SARS-CoV-2 virus; convalescent plasma—high titers of neutralizing antibodies against SARS-CoV-2 to experimentally treat several critical COVID-19 patients; tocilizumab—a humanized monoclonal antibody against the interleukin-6 receptor (IL6); favipiravir—an antiviral medication used to treat influenza and experimentally also COVID-19; hydroxycholochin (hydroxychloroquine)—suggested early in the pandemic as prevention or treatment method for COVID-19; remdesivir—a nucleotide analog prodrug indicated for treatment of COVID-19 disease in hospitalized patients.

**Figure 2 pathogens-09-01055-f002:**
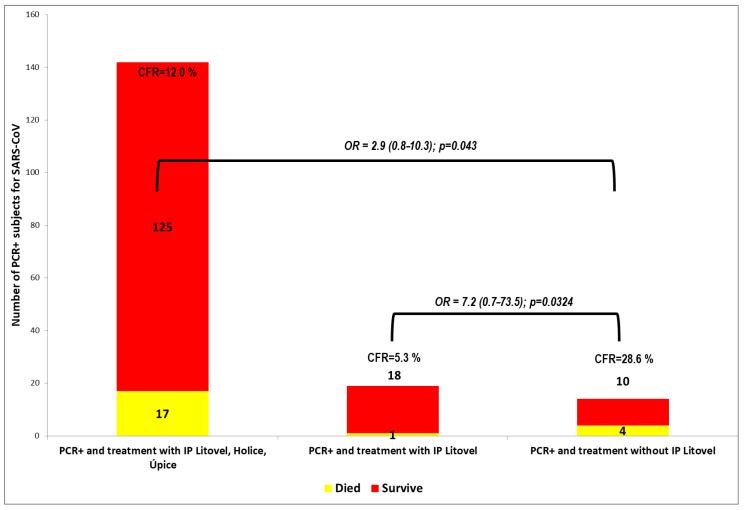
Comparison of the case-fatality rate (CFR) for COVID-19 of residents at three nursing homes (Litovel, Holice, and Úpice) that used IP, with the CFR for COVID-19 of the residents of the Litovel nursing home, some of whom used IP and some of whom did not use IP.

**Figure 3 pathogens-09-01055-f003:**
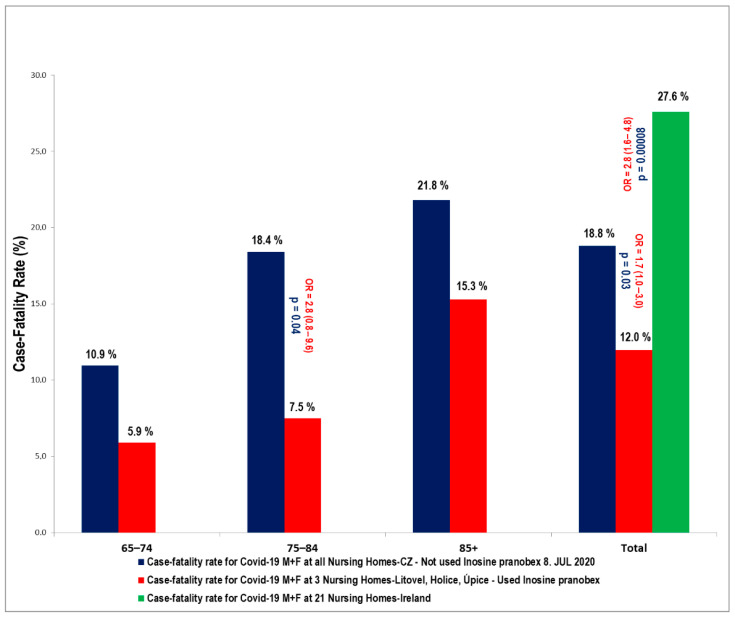
Comparisons of the CFR for COVID-19 of the residents of three selected nursing homes (Litovel, Holice, and Úpice) in the CZ (142 clients PCR+/17 died) that used IP, to the CFR for COVID-19 patients from all nursing homes (NH) in the CZ (415/78) until 8/8/2020, and to the CFR of 21 NH in Ireland (211/764).

**Table 1 pathogens-09-01055-t001:** Use of inosine pranobex in three nursing homes—Litovel, Sanatorium Topas Holice, and Beránek Úpice.

Nursing Homes	Capacity	PCR+	PCR+ with Inosine Pranobex	PCR+ with Inosine Pranobex (Patient Died)	PCR+ without Inosine Pranobex	PCR+ without Inosine Pranobex, (Patient Died)	PCR− without Inosine Pranobex
Litovel	56	33	19	1	14	4	23
Sanatorium Topas Holice	174	86	86	11	0	0	88 *
Beránek Úpice	71	37	37	5	0	0	34
Total	301	156	142	17	14	4	145

* All negative clients took inosine pranobex at a dose of 2 tablets once a day for 10 days when the epidemic first occurred in the nursing home (none became positive).

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
