# Peer review of "Inosine Pranobex Significantly Decreased the Case-Fatality Rate among PCR Positive Elderly with SARS-CoV-2 at Three Nursing Homes in the Czech Republic"

_pathogens, 2020, doi:10.3390/pathogens9121055_

Round 1

Reviewer 1 Report

The study entitled "Inosine pranobex significantly decreased the mortality rate among PCR positive elderly with SARS-CoV-2 virus at three Nursing Homes in the Czech Republic " by Jiri Beran et al. studies the effect of Inosine pranobex (IP) on COVID-19 patient. In an earlier trial, IP proved safe and effective for the treatment of individuals tested positive with acute respiratory viral infections. Authors in this report tested the effect of IP on the limited number of COVID-19 positive patients, especially with older individuals. The author claims that IP significantly reduced the mortality rate as compared to the control group. Although the trial is done on a very small group of individuals yet the manuscript is acceptable for publication after minor revision.

Major comments: The trial is incomprehensive and lacks control and placebo. However, a slightly significant result could also be considered with the disclaimer from authors. Hence, it is important to mention in the abstract, introduction, or discussion that these are preliminary results and that need further larger trials on COVID-19 patients.

Minor revision:
Recommended for “Case Report” or “Short Communication” rather than a “Full article”.
Figure 1: Minor revision: Figure 1 needs a better explanation in the legend. Describe all the components and symbols that have been mentioned in the figure and their significance.
Need to improve the quality of figures. Texts in figures are hardly readable.
Figure 2 & 3 could be combined?
Line 233: “Increases” to “increase”

Author Response

Reviewer-1:

The study entitled "Inosine pranobex significantly decreased the mortality rate among PCR positive elderly with SARS-CoV-2 virus at three Nursing Homes in the Czech Republic " by Jiri Beran et al. studies the effect of Inosine pranobex (IP) on COVID-19 patient. In an earlier trial, IP proved safe and effective for the treatment of individuals tested positive with acute respiratory viral infections. Authors in this report tested the effect of IP on the limited number of COVID-19 positive patients, especially with older individuals. The author claims that IP significantly reduced the mortality rate as compared to the control group. Although the trial is done on a very small group of individuals yet the manuscript is acceptable for publication after minor revision.

Major comments: The trial is incomprehensive and lacks control and placebo. However, a slightly significant result could also be considered with the disclaimer from authors. Hence, it is important to mention in the abstract, introduction, or discussion that these are preliminary results and that need further larger trials on COVID-19 patients.

Response:

This information was included into Abstract and Discussion:

Abstract line 31-32 as “Nevertheless, these are preliminary results that need larger-scale trials on COVID-19 patients.”

Discussion line 318-319: “Nevertheless, these are preliminary results that need confirmation using larger-scale trials on COVID-19 patients.”

Minor revision:

Recommended for “Case Report” or “Short Communication” rather than a “Full article”.

Response:

Thank you very much for the recommendation for “Case Report” or “Short Communication” rather than a “Full article”, but we want to stay at format “Full article”, because it is necessary to explain all aspects of inosine pranobex, NK cells and basic immunity processes to COVID-19 and short format of the manuscript doesn’t allow it.

Figure 1: Minor revision: Figure 1 needs a better explanation in the legend. Describe all the components and symbols that have been mentioned in the figure and their significance.

Response:

In the legend (lines 57-72) all the components and symbols that have been mentioned in the figure and their significance was described:

Lines 57-72: Legend: Red dot – Virus SARS-CoV-2, IP – inosine pranobex; NK cells – natural killers cells; CTL – cytotoxic T-lymphocyte (CD8+); APC - Antigen presenting cell; Naïve Th – naïve helper T lymphocyte (CD4+ cell); Th1 – type of helper cells lead to an increased cell-mediated immunity; Th2 - type of helper cells lead to a humoral immune (antibody) response; T-Cell - a type of lymphocyte, differentiate into helper; regulatory, or cytotoxic T cells or become memory T cells; B-Cell - B cells are involved in humoral immunity and differentiates into a plasma cell; Plasma Cell - short-lived antibody-producing cell derived from B-Cell; IgA, IgM, IgG – antibody classes of immunoglobulins; MF - Macrophages - specialized cells involved in the detection, phagocytosis and destruction of SARS-CoV-2 virus; Convalescent plasma - High titers of neutralizing antibodies against SARS-CoV-2 to experimentally treat several critical COVID-19 patients; Tocilizumab - a humanized monoclonal antibody against the Interleukin-6 receptor (IL6); Favipiravir - an antiviral medication used to treat influenza and experimentally also COVID-19; Hydroxycholochin (hydroxychloroquine) - suggested early in the pandemic as prevention or treatment method for COVID-19; Remdesivir - a nucleotide analog prodrug indicated for treatment of COVID-19 disease in hospitalized patients.

Need to improve the quality of figures. Texts in figures are hardly readable.

Response:
The quality of figures was improved and text is now readable.

Figure 2 & 3 could be combined?

Response:
Figures 2 & 3 were combined under new combined “Figure 2” lines 229-234

Line 233: “Increases” to “increase”

Response:
It has been corrected – now in line 275.

Reviewer 2 Report

Beran et al compare mortality rates in nursing homes (Czech Rep and Ireland) between Covid + patients either treated or untreated with Inosine Pranobex (IP). They conclude that IP treatment may reduce mortality rates in nursing home residents in all age groups tested.

Although this conclusion may be consistent with their data, there is insufficient information to assess this adequately.

Major points:

  1. Litovel NH study. Symptomatic Covid+ patients were given IP whereas asymptomatic patients were not. More untreated patients died but it is not clear what the cause of death was. Did they develop symptoms later or did they die of other causes?
  2. The difference in NH mortality rates between CZ and Ireland (untreated patients) is considerable. This suggests that comparison between nursing home is not particularly informative. Influence of demographics or other confounding factors has not been commented on in the paper.
  3. What other treatments did Covid+ NH residents receive in different NHs? For instance it is stated that Litovel patients received anti-Covid plasma. Were other treatments administered in the other nursing homes in the study?

These points should be adequately addressed for the study to meaningful.

Author Response

Reviewer-2:

Beran et al compare mortality rates in nursing homes (Czech Rep and Ireland) between Covid + patients either treated or untreated with Inosine Pranobex (IP). They conclude that IP treatment may reduce mortality rates in nursing home residents in all age groups tested.

Although this conclusion may be consistent with their data, there is insufficient information to assess this adequately.

Major points:

1. Litovel NH study. Symptomatic Covid+ patients were given IP whereas asymptomatic patients were not. More untreated patients died but it is not clear what the cause of death was. Did they develop symptoms later or did they die of other causes?

Response:

Section Recorded Case-Fatality Rate (line 186) was amended:

One patient treated with IP:

Lines 189-192: “The patient was an otherwise healthy 95-year-old man who was PCR+ for SARS-CoV-2, with no symptoms of COVID-19 from the date of his PCR test until the 6th day. He then, experienced acute onset of weakness and developed difficulty breathing (SpO2 92 %); treatment with 3 g of IP was initiated; death came suddenly 5 days later.”

Four patients no treated with IP

Lines 199-203: “The patients who died were very old (73, 90, 92, and 93 years). They did not develop typical respiratory symptoms of COVID-19. Instead, they typically experienced sudden onset weakness, low oxygen saturation (SpO2), or suspected micro-embolization. Importantly, all were polymorbid and none were treated with IP. Patient deaths were attributed to gastrointestinal problems (i.e., icterus, ileus, and gastroenteritis with severe electrolyte imbalances).”

2. The difference in NH mortality rates between CZ and Ireland (untreated patients) is considerable. This suggests that comparison between nursing home is not particularly informative. Influence of demographics or other confounding factors has not been commented on in the paper.

Response:

There aren’t any data in publication from NH in Ireland about resident’s demographic characteristics. It is obvious that higher age and male gender play important role. The discussion section was amended and three references were added:

Lines 423-429:

“24. Neill C, Sartaj M, Holcroft L, Hasan SS, Conway BR, Aldeyab MA. Surveillance study of asymptomatic and presymptomatic coronavirus disease 2019 (COVID-19) in care homes in Northern Ireland. Infect. Control Hosp.Epidemiol. 2020; 1-3.

25. Arons MM, Hatfield KM, Reddy SC et al. Presymptomatic SARS-CoV-2 Infections and Transmission in a Skilled Nursing Facility. N.Engl.J Med 2020; 382: 2081-2090.

26. Graham NSN, Junghans C, Downes R et al. SARS-CoV-2 infection, clinical features and outcome of COVID-19 in United Kingdom nursing homes. J Infect. 2020; 81: 411-419.”

Lines 290-297: “Recently, data with detailed demographics from 5 care homes in Northern Ireland were published [24]. In total, 388 individuals were tested (245 residents/87 positive, and 143 staff/10 positive). Most of the tested residents were women (72 %). Those testing positive had a mean age of 86.4 years (SD, 8.05) and CFR was 31 %. With regard to other international comparisons on NH COVID-19 infection rates and case-fatalities, results are consistent with a report on eighty-nine residents in a US facility [25], with an infection rate of 64 % and a CFR of 26 %, and a series from 394 residents in four UK NHs with an infection rate of 40 %, and a CFR of 26 % [26].”

Lines 301-304: “The average age of residents who tested PCR positive at the three NH (Litovel, Holice, Úpice) was 82.7, 84.9, and 85.5 years with median age 83.5, 86.9, and 86.0 years, respectively. Most of them were women (68.4 %, 72.1 %, 89.2 %, and for all three NH 76.1 %). This demographic data is very similar to other publications [24].”

The Subject section was also amended:
Lines 331-336: “The number of clients numbers, average age (AG), median age (MA), and percentage of women (PW) of PCR positive residents for the three selected homes were as follows (1) Litovel NH; 67, AG = 82.7 years, MA = 83,5 years, PW = 68.4 %, (2) Topas Holice NH; 174, AG = 84.9 years, MA = 86.9 years, PW = 72.1 %, and (3) Beranek Upice NH; 71, AG = 85.5 years, MA: 86.0 years, PW = 91.2 %, respectively. The PCR-testing for the SARS-CoV-2 virus was performed on all clients.”

3. What other treatments did Covid+ NH residents receive in different NHs? For instance it is stated that Litovel patients received anti-Covid plasma. Were other treatments administered in the other nursing homes in the study?

Response:

The treatment section was amended:

Lines 348-351: “At all three NH, PCR+ patients received symptomatic treatment and, if necessary also mineral supplementation, rehydration, and oxygenation. If residents developed signs of bacterial pneumonia (i.e., C-reactive protein > 70 mg/l, auscultation findings, fever, low SpO2), then antibiotic treatment was initiated.”

Lines 352-355: “Altogether there were 21 patients hospitalized at various infectious diseases departments. Except for one patient from the Litovel NH, none of them requested specific antiviral (anti-COVID-19) treatment during hospitalization (i.e., remdesivir or anti-COVID plasma); however, intensive antibiotic treatment, mineral supplementation, and rehydration was provided as needed.”

These points should be adequately addressed for the study to meaningful.

Round 2

Reviewer 2 Report

I am satisfied that the authors have adequately addressed the reviewers' points raised in response to the initial review.

Author Response

Dear Madam or Sir,

Thank you very much for your notes and recommendations.
In the revised manuscript all changes in response to your comments are marked using the “Track-Changes” tool.

Sincerely,

Professor Jiri Beran, MD
Director, Vaccination and Travel Medicine Centre
Tylovo nábÅ™eží 418/6, 500 02 Hradec Králové, Czech Republic
E-mail: jiri.beran@vakcinace.cz
www.vakcinace.cz